# Relaxed constraint and functional divergence of the progesterone receptor (PGR) in the human stem-lineage

Mirna Marinić[1], Vincent J. Lynch[2]*

1 Department of Organismal Biology and Anatomy, University of Chicago, Chicago, IL, United States of America, 2 Department of Biological Sciences, University at Buffalo, SUNY, Buffalo, NY, United States of America

* vjlynch@buffalo.edu

**Data Availability Statement:** All relevant data are within the manuscript and its Supporting Information files.

**Funding:** The funders had no role in study design, data collection and analysis, decision to publish, or

## Abstract

The steroid hormone progesterone, acting through the progesterone receptor (PR), a ligand-activated DNA-binding transcription factor, plays an essential role in regulating nearly every aspect of female reproductive biology. While many reproductive traits regulated by PR are conserved in mammals, Catarrhine primates evolved several derived traits including spontaneous decidualization, menstruation, and a divergent (and unknown) parturition signal, suggesting that PR may also have evolved divergent functions in Catarrhines. There is conflicting evidence, however, whether the progesterone receptor gene (*PGR*) was positively selected in the human lineage. Here we show that *PGR* evolved rapidly in the human stem-lineage (as well as other Catarrhine primates), which likely reflects an episode of relaxed selection intensity rather than positive selection. Coincident with the episode of relaxed selection intensity, ancestral sequence resurrection and functional tests indicate that the major human PR isoforms (PR-A and PR-B) evolved divergent functions in the human stem-lineage. These results suggest that the regulation of progesterone signaling by PR-A and PR-B may also have diverged in the human lineage and that non-human animal models of progesterone signaling may not faithfully recapitulate human biology.

## Author summary

The hormone progesterone regulates nearly every aspect of female reproductive biology, including when implantation will occur, the timing of the reproductive cycle, as well as the maintenance and cessation of pregnancy at term. These actions are mediated by the progesterone receptor, a transcription factor that binds progesterone and controls the expression of thousands of genes related to reproductive biology. Remarkably, previous studies have suggested that the progesterone receptor evolved rapidly in humans, with signatures of positive selection. However, using a large dataset of mammalian progesterone receptor genes, we found that selection on the progesterone receptor has actually been weakened in humans, with no evidence for positive selection. To explore if this human-specific episode of relaxed selection is associated with functional changes, we resurrected

preparation of the manuscript. This study was supported by a grant from the March of Dimes (March of Dimes Prematurity Research Center to principal investigator VJL) and a Burroughs Wellcome Fund Preterm Birth Initiative grant (1013760, to principal investigator VJL).

**Competing interests:** The authors have declared that no competing interests exist.

ancestral forms of the progesterone receptor and tested their ability to regulate a target gene. We found that the human progesterone receptor forms have changed in function, suggesting the actions regulated by progesterone may also be different in humans. Our results suggest caution in attempting to apply findings from animal models to progesterone biology of humans.

## Introduction

The steroid hormone progesterone plays a central role in female reproduction. In Eutherian (Placental) mammals, for example, progesterone regulates the timing of the reproductive cycle, ovulation, decidualization, implantation and the maintenance of pregnancy, myometrial quiescence, and the cessation of pregnancy at parturition [1–3]. Many of the biological actions of progesterone are mediated through the progesterone receptor (PR), which acts as a ligand-activated DNA-binding transcription factor [1,2,4]. The progesterone receptor gene (*PGR*) encodes two well-characterized isoforms (PR-A and PR-B) that are transcribed from distinct promoters and utilize different translation start sites in the first exon, but are identical except for a 165 amino acid trans-activation domain in the amino terminus of PR-B [5]. Consistent with the structural differences between PR-A and PR-B, previous studies have shown that PR-B is a stronger trans-activator of progesterone responsive genes than PR-A, whereas PR-A acts as a dominant trans-repressor of PR-B mediated trans-activation [5–9]. These opposing transcriptional activities result from different post-translational modifications and cofactor interactions. For example, PR-A does not efficiently interact with co-activators but strongly binds the co-repressor SMRT, allowing it to function as a dominant trans-repressor [6–8].

While female reproductive traits, particularly those regulated by PR, are generally well conserved across Eutherian [10] and Therian mammals [11–13], the genes that underlie reproductive traits in a wide range of taxa can evolve rapidly, often under the influence of positive selection [14–16]. Two primate lineages differ dramatically in some otherwise conserved female reproductive traits. Catarrhine primates have evolved spontaneous differentiation (decidualization) of endometrial stromal fibroblasts (ESFs) into decidual stromal cells (DSCs) under the direct action of progesterone [17,18]; menstruation [19,20]; and a divergent (and unknown) parturition signal [21,22], although uterine inflammation likely plays an important role in parturition [23]. In addition, humans have evolved permanently enlarged breasts [24,25], concealed ovulation [26,27], and longer pregnancy and labor compared to other primates [28–30]. These data suggest that PR may have rapidly evolved in humans and Catarrhines [here we define "rapid evolution" as a lineage-specific increase in the relative rate of nucleotide substitution compared to other lineages *sensu* Pollard *et al.* (2006) [31]].

There is conflicting evidence whether *PGR* was positively selected in the human lineage. *PGR* was ranked 8th in a genome-wide scan for positive selection in human-chimp-mouse gene trios [32] and 29th among genes with the strongest statistical evidence of positive selection in a pair-wise genome-wide scan for positively selected genes in human and chimpanzee genomes [33]. In contrast, evidence of positive selection on *PGR* was not detected in the human lineage in analyses of human-chimp-macaque gene trios [34], human-chimp-mouse-rat-dog orthologs [35], a dataset including seven primates [36], or a dataset including nine primates [37]. A study of human-chimp-macaque-mouse-rat-dog orthologs found evidence for a positively selected class of sites in the human lineage (4.45%, $d_N/d_S = 3.15$), but the results were not statistically significant (LRT $P = 0.46$) [38]. The most comprehensive analysis of *PGR* evolution thus far, which included 14 primates and four outgroups [29], found evidence for an

episode of rapid evolution in the human lineage. However, this study did not explicitly test whether the $d_N/d_S$ rate was significantly greater than 1, and while a branch-sites test identified a proportion of sites as potentially positively selected (17%, $d_N/d_S$ = 5.5), the null hypothesis could not be rejected (LRT $P$ = 0.22).

To resolve these conflicting data, we assembled a dataset of 119 Eutherian *PGRs*, including species and sub-species from each primate lineage, as well as modern and archaic humans (Neanderthal and Denisovan), and used a suite of maximum likelihood-based methods to characterize the strength and direction of selection acting on *PGR*. We found that *PGR* evolved rapidly in the human lineage, however, there was little evidence it was driven by an episode of positive selection. Rather, the rate acceleration was consistent with a relaxation in the intensity of purifying selection, which likely reflects a long-term trend in Catarrhine primates. To test whether the episode of relaxed constraint occurred coincident with a change in the function of the human PR, we resurrected the ancestral human (AncHuman) and human/chimp (AncHominini) PR-A and PR-B isoforms and tested their ability to trans-activate reporter gene expression from the decidual *Prolactin* promoter (dPRL-332), a well-characterized progesterone responsive element [39–43]. We found pronounced functional differences between AncHuman and AncHominini PR isoforms, particularly in the ability of PR-A to trans-repress PR-B. These data suggest that an episode of relaxed purifying selection altered the function of the human PR, which may have impacted female reproductive biology.

## Results

### No evidence of positive selection in the human stem-lineage

We assembled a dataset of 119 Eutherian *PGRs*, including representatives of each major primate lineage, to characterize the strength and direction of selection generally acting on these genes and explicitly test for an episode of positive selection in the human-lineage using the species phylogeny shown in **Fig 1**. We first compared two likelihood models using HyPhy [44]: a one-ratio model in which all lineages have a single $d_N/d_S$ ratio (ω), and a two-ratio model in which the $d_N/d_S$ ratio in the human stem-lineage ($ω_1$) was estimated separately from the $d_N/d_S$ ratio in all other lineages ($ω_2$). We allowed for variable $d_N$ and $d_S$ cross sites with 3 rate classes in both models. The two ω model was a significantly better fit to the data than the single ω model, with $ω_2$ = 2.67 (95% CI = 1.15–5.18; LRT = 8.91, $P$ = 2.83×10$^{-3}$), however, the $d_N/d_S$ ratio in the human stem-lineage was not significantly different than 1 (LRT = 1.15, $P$ = 0.28). To test if there was a class of sites with ω>1 in the human stem-lineage, we used an adaptive branch-site random effects likelihood (aBSREL) model [45] that infers the optimal number of site classes and the ω of each site class for each lineage. The aBSREL model inferred a class of sites in the human stem-lineage with ω = 2.80, but ω was not significantly different than 1 ($P$ = 0.44). We also used the branch-site unrestricted statistical test for episodic diversification (BUSTED) model [46], which can detect positive selection on a subset of branches and at a subset of sites, and again did not infer either episodic diversifying selection ($P$ = 0.57), or support for positive selection on individual substitutions in the human stem-lineage (evidence ratios ~2). In contrast, aBSREL and BUSTED did identify evidence of positive selection in non-primate lineages (**S1 Table**).

### No evidence of positive selection at sites with human-specific substitutions

The methods used above can detect positive selection acting on *a priori* defined lineages (two ω-ratio model) and classes of sites across all lineages (aBSREL and BUSTED), but cannot detect positive selection acting on specific sites, alignment-wide. To test for pervasive positive and negative selection at individual amino acid sites, we used two distinct models: Fixed

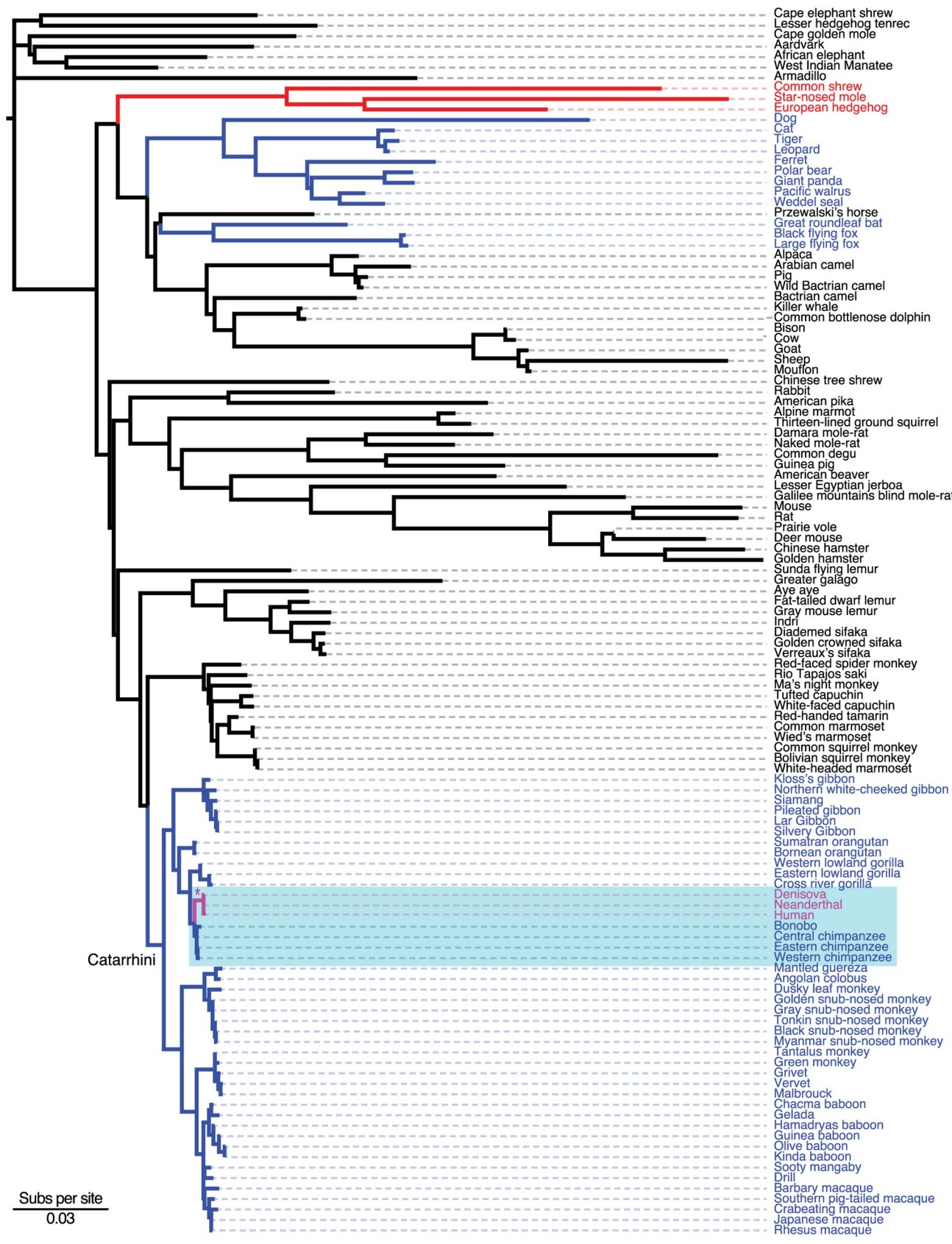

**Fig 1. Phylogeny of species used to characterize the strength and direction of selection on *PGR* genes.** Branch lengths are drawn proportional to the number of substitutions per site [Subs per site, see inset scale bar] under the GTR model. Substitutions per site are a measure of sequence divergence–long branches are more diverged than shorter branches. The human stem-lineage, modern human, Denisovan, and Neanderthal lineages are shown in magenta; the Hominini clade is shown in cyan. Asterisk (*) indicates the human stem-lineage, which has an $\omega = 1.06$ and $k = 0.00$ (LRT = 5.01, $P = 0.01$) under the RELAX *a priori* model. Branches are colored black if k is not significantly different than 1, red if k is significantly greater than 1 and blue if k is significantly less than 1 in the *ad hoc* RELAX analyses.

Effects Likelihood (FEL) and Single-Likelihood Ancestor Counting (SLAC) [47]. FEL, which uses a maximum-likelihood approach to infer $d_N$ and $d_S$ substitution rates on a per-site basis, found five sites with evidence of pervasive positive selection and 583 sites with evidence of purifying selection (**S1 Table**). Similarly the SLAC model, which combines maximum-likelihood and counting approaches to infer $d_N$ and $d_S$, showed evidence for positive selection at four sites and purifying selection at 531 sites (**S1 Table**). Neither method inferred positive selection at sites with human-specific amino acid changes (**Fig 2C**).

Both FEL and SLAC assume that the selection pressure for each site is constant across the entire phylogeny, thus sites are not allowed to switch between positive, negative, and neutral rate categories. This assumption, however, may not be realistic when evolution is episodic rather than pervasive. Therefore, we use the Fast Unconstrained Bayesian AppRoximation (FUBAR) [48] and the Mixed Effects Model of Evolution (MEME) methods [49] to test for episodic positive selection at amino acid sites. In addition to detecting episodic selection, FUBAR may have more power than FEL when positive selection is present but weak, i.e. low values of $\omega > 1$ [48]. FUBAR identified a single site with evidence of positive selection and 689 sites with evidence of purifying selection (**S1 Table**), whereas MEME found evidence of positive selection at 17 sites (**S1 Table**). Of the eight human-specific substitutions (see below), all occurred at sites that are inferred by FUBAR or MEME to evolve under negative selection or neutrally alignment-wide (**Tables 1–4; Fig 2C**). These data suggest that while *PGR* may have experienced positive selection at some sites in some Eutherian lineages, the episode of rapid evolution in the human stem-lineage is unlikely to result from either pervasive or episodic positive selection.

## Relaxed purifying selection on human and primate *PGRs*

To test if the episode of rapid evolution in the human stem-lineage may result from a relaxation in the intensity of selection (both positive and negative), rather than an episode of positive selection, we used a variant of the aBSREL method (RELAX) that explicitly includes a selection intensity parameter (k) [50]. A significant $k > 1$ indicates that selection strength has been intensified, whereas a significant $k < 1$ indicates that the strength of selection (both positive and negative) has been relaxed. RELAX inferred that the baseline rate distribution for branch-site combinations was $\omega_1 = 0.00$ (59.41% of sites), $\omega_2 = 0.07$ (34.78% of sites), and $\omega_3 = 1.26$ (5.81% of sites), while the branch-level relaxation or intensification (k) parameter distribution had a mean of 1.17, median of 0.36, and 95% of the weight within the range of 0.03–6.62. Thus, the $\omega_3$ class of sites was inferred to be evolving near neutrality ($\omega = 1$), alignment-wide. Consistent with an episode of relaxed selection, RELAX inferred $\omega = 1.06$ and $k = 0.00$ (LRT = 5.01, $P = 0.01$) in the human stem-lineage (**Fig 1**).

To explore if the relaxation of constraint was specific to the human lineage or reflects a more widespread pattern, we also tested for relaxed purifying selection in other clades. We found evidence for relaxation in *Homo* (modern and archaic humans; $k = 0.00$, $P = 0.012$, LRT = 6.38), Hominoidea (apes; $k = 0.68$, $P = 0.09$, LRT = 2.89), Cercopithecidae ('Old World monkeys'; $k = 0.43$, $P = 7.28 \times 10^{-6}$, LRT = 20.12), Cercopithecinae ($k = 0.44$, $P = 6.60 \times 10^{-4}$, LRT = 11.60), Colobinae ($k = 0.32$, $P = 1.25 \times 10^{-3}$, LRT = 10.41), and Catarrhini ('Old World

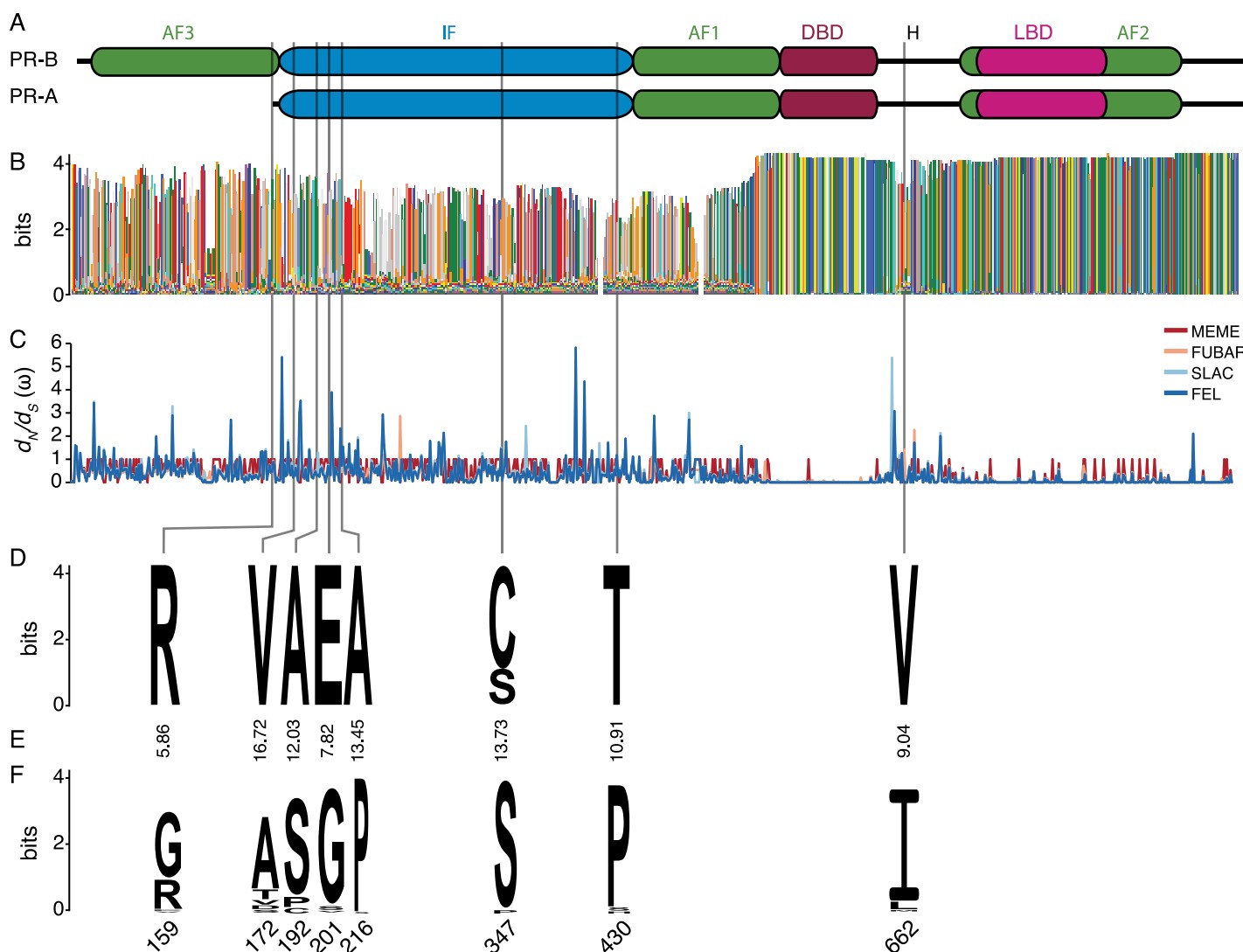

**Fig 2. Distribution of human-specific amino acid substitutions in PGR.** (A) Functional domains of the progesterone receptor isoforms A (PR-A) and B (PR-B). AF1-3, activation function domains 1–3. IF, inhibitory function domain. DBD, DNA-binding domain. H, hinge region. LBD, ligand-binding domain. (B) Conservation of PGR across 119 Eutherian mammals shown as a logo [colors are arbitrary]. Bits are a measure of randomness; random sequences have a low bit score while non-random [conserved] sequences have a high bit score [max 4.2 for amino acid sequences]. (C) Site-specific $d_N/d_S$ ratios (ω) inferred by MEME (red), FUBAR (pink), SLAC (light blue), and FEL (dark blue). (D) Logo showing conservation of human-specific substitutions within modern and archaic humans. (E) CADD scores for human-specific substitutions. (F) Logo showing conservation at sites with human-specific substitutions in non-human mammals.

monkeys' and apes; k = 0.43, P = 1.03×10⁻⁵, LRT = 19.39), but not Platyrrhini ('New World monkeys'; k = 0.95, P = 0.681, LRT = 0.17), Strepsirrhini (lemurs and lorises; k = 0.94, P = 0.37, LRT = 0.80), or non-primates (k = 2.41, P = 3.06×10⁻⁸, LRT = 30.67). Thus, we conclude the rate acceleration in the human stem-lineage most likely results from a long-term trend of relaxed purifying selection on Catarrhine primate *PGRs*, rather than an episode of positive selection.

## No evidence for positive selection in Hominini

Chen *et al.* (2008) [29] found the strongest evidence for positive selection when the human and chimpanzee lineages were analyzed together as a single clade (Hominini). Therefore, we

**Table 1. Fixed Effects Model (FEL) results for sites with human-specific amino acid substitutions.** α, synonymous substitution rate. β, non-synonymous substitution rate. ω, $d_N/d_S$ ratio. α = β, rate estimate under the neutral model. LRT, likelihood ratio test statistic for α = β versus α < β. Branch length, the total length of branches contributing to the inference at this site (used to scale $d_N$ and $d_S$).

| Site | α | β | ω | α = β | LRT | P-value | Branch length |
|---|---|---|---|---|---|---|---|
| 159 | 1.42 | 0.63 | 0.44 | 0.85 | 2.80 | 0.10 | 7.37 |
| 172 | 0.86 | 0.66 | 0.77 | 0.72 | 0.24 | 0.62 | 6.18 |
| 192 | 0.27 | 0.38 | 1.40 | 0.35 | 0.17 | 0.68 | 3.02 |
| 201 | 0.54 | 0.30 | 0.56 | 0.38 | 0.69 | 0.41 | 3.21 |
| 216 | 5.15 | 0.21 | 0.04 | 0.63 | 12.36 | 0.00 | 13.97 |
| 347 | 2.76 | 0.25 | 0.09 | 0.86 | 16.41 | 0.00 | 8.33 |
| 430 | 1.24 | 0.41 | 0.33 | 0.64 | 3.11 | 0.08 | 5.58 |
| 662 | 0.14 | 0.43 | 3.08 | 0.35 | 1.40 | 0.24 | 2.98 |

also tested for evidence of positive and relaxed selection in the Hominini clade (Fig 1). We again found that a two-ratio model, in which the $d_N/d_S$ ratio in the Hominini clade ($\omega_1$) was estimated separately from all other lineages ($\omega_2$), was a significantly better fit to the data than the single ω model, with $\omega_2$ = 2.19 (95% CI = 1.1–5.99; LRT = 10.10, $P = 6.00 \times 10^{-3}$), but the $d_N/d_S$ ratio for the Hominini clade was not significantly different than 1 (LRT = 1.55, $P$ = 0.46). Similarly, neither aBSREL nor BUSTED inferred evidence for positive selection in the Hominini clade (S1 Table). In contrast, the RELAX test for selection relaxation (k = 0.08) was significant ($P = 6.00 \times 10^{-3}$, LRT = 7.55; S1 Table). Thus, the rate acceleration in the Hominini clade also likely reflects a long-term trend of relaxed purifying selection on Catarrhine primate *PGRs*, rather than an episode of positive selection in the Hominini.

## Human-specific amino acid substitutions are predicted to be deleterious

Similar to previous studies [29], we identified eight human-specific amino acid substitutions (G159R, A172V, S192A, G201E, P216A, S347C, P430T, and I662V), six of which are located in the inhibitory function (IF) domain, one in the activation function 3 (AF3) domain, and one in the hinge region (H) (Fig 2A). While three human-specific substitutions are fixed for the derived amino acid, ancestral variants are segregating at very low frequencies at sites 172 (rs376101426, ancestral allele frequency = $1.37 \times 10^{-5}$), 201 (rs748082098, ancestral allele frequency = $4.33 \times 10^{-6}$), 347 (rs11571147, ancestral allele frequency = $9.08 \times 10^{-3}$), 430 (rs1396844023, ancestral allele frequency = $6.08 \times 10^{-6}$), and 662 (rs150584881, ancestral allele

**Table 2. Single-Likelihood Ancestor Counting (SLAC) results for sites with human-specific amino acid substitutions.** ES, expected synonymous sites. EN, expected non-synonymous sites. S, inferred synonymous substitutions. N, inferred non-synonymous substitutions. P[S], expected proportion of synonymous sites. $d_S$, inferred synonymous substitution rate. $d_N$, inferred non-synonymous substitution rate. P [$d_N/d_S > 1$], binomial probability that S is no greater than the observed value, with $P_s$ probability of success. P ($d_N/d_S < 1$), binomial probability that S is no less than the observed value, with $P_s$ probability of success. Total branch length, the total length of branches contributing to the inference at this site (used to scale $d_N$ and $d_S$).

| Site | ES | EN | S | N | P[S] | $d_S$ | $d_N$ | $d_N/d_S$ | P [$d_N/d_S > 1$] | P [$d_N/d_S < 1$] | Total branch length |
|---|---|---|---|---|---|---|---|---|---|---|---|
| 159 | 0.97 | 1.99 | 9 | 10 | 0.33 | 9.29 | 5.01 | 0.54 | 0.94 | 0.13 | 3.71 |
| 172 | 0.95 | 1.92 | 6 | 10 | 0.33 | 6.31 | 5.21 | 0.83 | 0.74 | 0.45 | 3.85 |
| 192 | 0.95 | 1.97 | 2 | 6 | 0.33 | 2.10 | 3.04 | 1.45 | 0.49 | 0.79 | 3.76 |
| 201 | 0.98 | 1.94 | 4 | 5 | 0.33 | 4.10 | 2.57 | 0.63 | 0.85 | 0.35 | 3.76 |
| 216 | 1.00 | 2.00 | 4 | 2 | 0.33 | 3.99 | 1.00 | 0.25 | 0.98 | 0.10 | 3.67 |
| 347 | 0.87 | 1.68 | 11 | 3 | 0.34 | 12.60 | 1.79 | 0.14 | 1.00 | 0.00 | 3.59 |
| 430 | 0.90 | 1.84 | 6 | 5 | 0.33 | 6.64 | 2.71 | 0.41 | 0.96 | 0.12 | 3.46 |
| 662 | 0.79 | 2.22 | 1 | 8 | 0.26 | 1.27 | 3.60 | 2.83 | 0.27 | 0.93 | 3.88 |

**Table 3. Mixed Effects Model of Evolution (MEME) results for sites with human-specific amino acid substitutions.** α, synonymous substitution rate. β-, non-synonymous substitution rate for the negative/neutral evolution component. p-, mixture distribution weight allocated to β- (i.e., the proportion of the tree neutrally of or under negative selection). β+, non-synonymous substitution rate for the positive selection/neutral component. p+, mixture distribution weight allocated to β+ (i.e., the proportion of the tree neutrally of or under positive selection). LRT, likelihood ratio test statistic for episodic diversification(i.e., p+ > 0 and β+ > α). *P*-value, asymptotic *P*-value for episodic diversification (i.e., p+ > 0 and β+ > α). # branches under selection, an estimate for how many branches may been under selection at this site. Total branch length, the total length of branches contributing to the inference at this site (used to scale $d_N$ and $d_S$).

| Site | α | β - | p- | β + | p+ | LRT | *P*-value | # branches under selection | Total branch length |
|------|------|------|------|------|------|------|------|------|------|
| 159 | 1.48 | 1.48 | 0.47 | 0.00 | 0.53 | 0.00 | 0.67 | 0 | 7.97 |
| 172 | 0.86 | 0.79 | 0.67 | 0.4 | 0.33 | 0.00 | 0.67 | 0 | 6.21 |
| 192 | 0.27 | 0.27 | 0.04 | 0.38 | 0.96 | 0.17 | 0.54 | 0 | 3.03 |
| 201 | 0.69 | 0.69 | 0.37 | 0.00 | 0.63 | 0.00 | 0.67 | 0 | 3.3 |
| 216 | 5.82 | 5.82 | 0.06 | 0.00 | 0.94 | 0.00 | 0.67 | 0 | 16.32 |
| 347 | 2.75 | 1.61 | 0.16 | 0.03 | 0.84 | 0.00 | 0.67 | 0 | 8.56 |
| 430 | 1.24 | 0.41 | 0.89 | 0.37 | 0.11 | 0.00 | 0.67 | 0 | 5.58 |
| 662 | 0.14 | 0.14 | 0.00 | 0.43 | 1.00 | 1.40 | 0.25 | 0 | 2.98 |

frequency = $9.59\times10^{-5}$). Similarly, nearly all human-specific amino acid substitutions are fixed for the derived allele in archaic humans, but Neanderthal has 347C whereas Denisovan has 347S (**Fig 2**). We next used Combined Annotation-Dependent Depletion (CADD) [51,52] to predict the deleteriousness of each human-specific amino acid substitution and found that most are predicted to be deleterious, and occur at sites that are not under episodic or pervasive positive selection, alignment-wide (**Fig 2**).

## Functional divergence of the human progesterone receptor

To determine if human-specific substitutions have functional consequences, we reconstructed the sequences of the ancestral human (AncHuman) and ancestral human-chimpanzee (AncHominini) PRs (**Fig 3A**), synthesized the PR-A and PR-B isoforms, and cloned them into a mammalian expression vector. Next, we transiently transfected mouse embryonic fibroblasts (MEFs]), which do not express *Pgr*, with either AncHuman PR-A, AncHuman PR-B, AncHuman PR-A/PR-B, AncHominini PR-A, AncHominini PR-B, or AncHominini PR-A/PR-B expression vectors and a reporter vector that drives luciferase expression from the decidual *Prolactin* promoter (dPRL-332). The dPRL-332 promoter is a very well-characterized progesterone responsive regulatory element that is bound by PR-A and PR-B and directs the expression of *Prolactin* in decidual stromal cells (**Fig 3B**) [39–43]. The AncHuman PR-A and AncHominini PR-A isoforms weakly trans-activated luciferase expression from the dPRL-332 promoter, both PR-B isoforms strongly trans-activated luciferase expression, and both PR-A

**Table 4. Fast Unconstrained Bayesian AppRoximation (FUBAR) results for site with human-specific amino acid changes.** α, mean posterior synonymous substitution rate. β, mean posterior non-synonymous substitution rate. Mean posterior β-α. P[α > β], posterior probability of negative selection. P[α < β], posterior probability of positive selection. BF[α < β], empirical Bayes factor for positive selection. PSRF, potential scale reduction factor (a measure of MCMC mixing). Neff, effective sample size.

| Site | α | β | β—α | P[α > β] | P [α < β] | BF[α < β] | PSRF | Neff |
|------|------|------|------|------|------|------|------|------|
| 159 | 2.77 | 1.15 | -1.62 | 0.98 | 0.00 | 0.02 | 1.02 | 112.62 |
| 172 | 2.04 | 1.10 | -0.95 | 0.75 | 0.09 | 0.66 | 1.03 | 83.33 |
| 192 | 0.75 | 0.70 | -0.05 | 0.49 | 0.41 | 4.55 | 1.01 | 262.30 |
| 201 | 1.09 | 0.64 | -0.45 | 0.79 | 0.13 | 0.97 | 1.00 | 267.69 |
| 216 | 5.22 | 0.58 | -4.63 | 1.00 | 0.00 | 0.01 | 1.02 | 102.74 |
| 347 | 3.59 | 0.60 | -2.99 | 1.00 | 0.00 | 0.00 | 1.02 | 99.99 |
| 430 | 2.53 | 0.87 | -1.66 | 0.97 | 0.01 | 0.08 | 1.02 | 92.42 |
| 662 | 0.57 | 0.77 | 0.20 | 0.26 | 0.66 | 12.72 | 1.01 | 180.06 |

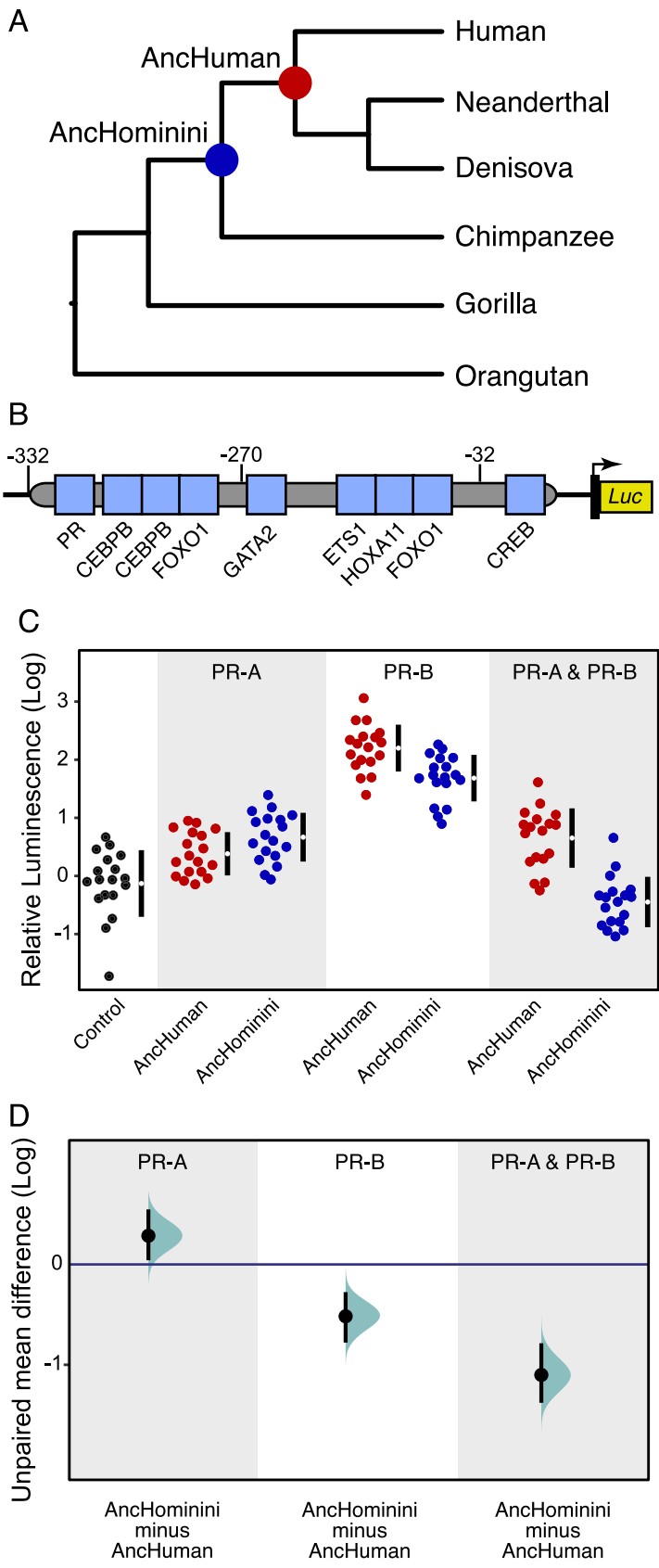

**Fig 3. Functional divergence of the human progesterone receptor.** (A) Hominini (African ape) phylogeny, nodes with ancestral sequence reconstructions are indicated. (B) The dPRL-332 luciferase reporter vector with the location of experimentally verified transcription factor binding sites. (See main text for references). (C) Strip chart showing relative luminescence values, standardized to negative control (log scale). Mean is depicted as a white dot; 95% confidence intervals are indicated by the ends of the black vertical error bars. (D) The mean difference in luminescence of AncHuman against AncHominini shown as a Cumming estimation plot (log scale). Mean differences are plotted as bootstrap sampling distributions ($n$ = 5,000). Each mean difference is depicted as a dot, 95% confidence intervals are indicated by the ends of the black vertical error bars. The AncHuman PR-A was a weaker trans-activator than the AncHominini PR-A (Mann-Whitney $P$ = 0.035), while the AncHuman PR-B was a stronger trans-activator than the AncHominini PR-B (Mann-Whitney $P$ = 9.46×10$^{-4}$). The ability of AncHuman PR-A to repress AncHuman PR-B was weaker than AncHominini PR-A on AncHominini PR-B (Mann-Whitney $P$ = 3.06×10$^{-6}$).

isoforms trans-repressed trans-activated luciferase expression by PR-B (**Fig 3C**). The AncHuman PR-A was a weaker trans-activator than the AncHominini PR-A with a log mean difference of 0.548 (95% CI: 0.099–1.03, Mann-Whitney $P$ = 0.035), while the AncHuman PR-B was a stronger trans-activator than the AncHominini PR-B with a log mean difference of -0.517 (95% CI: -0.779 –-0.276, Mann-Whitney $P$ = 9.46×10$^{-4}$). Additionally, the ability of AncHuman PR-A to repress AncHuman PR-B was weaker than AncHominini PR-A on AncHominini PR-B with a log mean difference of -1.10 (95% CI: -1.38 –-0.785, Mann-Whitney $P$ = 3.06×10$^{-6}$) (**Fig 3D**).

## Discussion

### Positive selection, relaxation, or both?

Numerous previous studies have conflicted over whether *PGR* was positively selected in the human lineage [29,32–36]. These studies had limited taxon sampling (particularly within primates) and used models designed to detect positive selection rather than explicitly test evidence for relaxed purifying selection [50]. This may have reduced their power to differentiate an episode of positive selection from a relaxation in the intensity. Our results from multiple methods indicate that *PGR* evolved under relatively strong purifying selection in most lineages and sites and suggest that *PGR* experienced an episode of rapid evolution in the human stem-lineage. However, no method found that an inference of positive selection (ω>1) was supported, whereas several methods found strong support for an episode of relaxed constraint (ω = 1) on at least some sites in the human stem-lineage. Our results also suggest that the intensity of purifying selection acting on Catarrhine primate *PGRs* has been relaxed, implying that the episode of relaxed selection intensity in the human stem-lineage (and within Hominini) reflects a wider trend in Catarrhines.

These observations are particularly interesting given the finding of substantial differentiation at the *PGR* locus between human populations, suggesting a recent episode of positive selection in East Asians [53]. Li *et al*. (2018) found that derived single nucleotide variants near the *PGR* locus at high frequency in Han Chinese in Beijing, China (CHB) were associated with *PGR* expression in the ovary, but not other progesterone responsive tissues such as the uterus, vagina, or breast. These high frequency derived variants were also associated by a GWAS with an increased risk of early spontaneous preterm birth (≤ 32 gestational weeks; early sPTB) in African Americans, suggesting they predispose to early sPTB either generally, or in populations other than CHB. In contrast, Li et al. observed that derived alleles at the *PGR* locus that are at high frequencies in multiple human populations were associated with a reduced risk for early sPTB in the same GWAS cohort. Thus, while there is a trend toward relaxed selection intensity on *PGR* in humans and other Catarrhine primates, within modern humans directional selection may be acting to reduce early sPTB risk.

## Functional divergence of human progesterone receptor

Regardless of the statistical evidence supporting positive selection or relaxed constraint, we found that human PR-A and PR-B have diverged in function compared to the human-chimp ancestor. The trans-activation strength of the AncHuman PR-A is slightly weaker than the AncHominini PR-A, whereas the AncHuman PR-B more strongly trans-activates luciferase expression from the dPRL-332 promoter than the AncHominini PR-B. In addition to acting as a ligand-activated transcriptional activator, PR-A also has the ability to trans-repress the transcriptional activity of PR-B and other nuclear receptors, such as estrogen and glucocorticoid receptors, through its inhibitory function domain [6]. The most striking difference between the AncHuman and AncHominini isoforms is in the trans-repressive strength of PR-A on PR-B – the AncHominini PR-A can almost completely repress trans-activation by AncHominini PR-B, while the AncHuman PR-A's capacity to inhibit trans-activation by AncHuman PR-B is weakened compared to AncHominini PR-A. Thus, PR-B can still function as a trans-activator in the presence of stoichiometric ratios (1:1) of PR-A. These results are consistent with a previous study which compared the trans-activation and trans-repressive strengths of modern human and mouse PR-A and PR-B and found that the human PR-A was a weaker trans-repressor of PR-B than the mouse PR-A [54].

## It is tempting to speculate

Identifying the cause(s) of the relaxed selection intensity acting on PGR in Catarrhine primates and the human stem-lineage is challenging because progesterone signaling has many functions and many progesterone-dependent traits are phylogenetically associated with Catarrhines, any one or none of which may underlie relaxed selection. Given the essential role progesterone plays in female reproduction [1–3], however, it is reasonable to conclude that relaxed selection intensity might be related to some aspect of female reproductive biology. Thus, it is interesting to note that Catarrhines have evolved a divergent (and unknown) parturition signal. Unlike nearly every other Eutherian mammal, parturition in Catarrhine primates is not associated with a rapid decline in systemic progesterone concentrations, instead progesterone concentration remains relatively stable throughout pregnancy in Cercopithecidae ('Old World monkeys') or continues to increase until parturition in apes [55]. These data suggest that systematically high progesterone levels throughout pregnancy may have reduced the strength of purifying selection acting on *PGR* in Catarrhines, perhaps because these continually high concentrations compensated for the fixation of deleterious amino acid substitutions.

A more interesting question is whether the enhanced trans-activation ability of PR-B and the weakened trans-repressive strength of PR-A in the human lineage has functional consequences? In the absence of systemic progesterone withdrawal as a mechanism to induce parturition, it has been proposed that progesterone is "functionally" withdrawn in Catarrhine primates [21,22]. Many mechanisms have been proposed to underlie functional progesterone withdrawal, including local breakdown of progesterone in the endometrium and myometrium [56,57], sequestration of progesterone in the plasma by carrier proteins [58–62], a decrease in the levels of PR co-activators at term [63], functional estrogen activation [64], and inflammation resulting in NF-κB-mediated PR repression [65]. A particularly popular hypothesis is that shifts in the abundance of PR-A and PR-B induce functional progesterone withdrawal–when PR-B is the dominant isoform progesterone signaling is active, whereas when PR-A is the dominant isoform progesterone signaling through PR-B is inhibited [66–69]. Consistent with the isoform switching hypothesis, the PR-A/PR-B ratio in the myometrium transitions towards the end of pregnancy with PR-A increasing in abundance until parturition in humans [67], macaques [66], and mouse [70]. For example, PR-A/PR-B protein ratio in the human

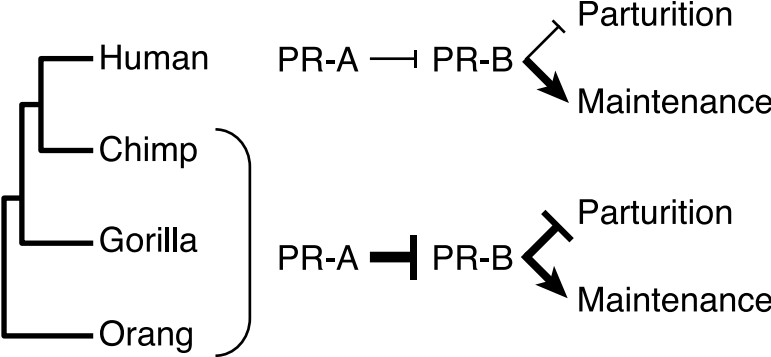

**Fig 4. Model of progesterone receptor action under the isoform switching hypothesis.** Progesterone signaling through PR-B promotes the maintenance of pregnancy and inhibits parturition, while PR-A induces the cessation of pregnancy and parturition by repressing the actions of PR-B. PR-B evolved stronger trans-activation ability in the human lineage, while the ability of PR-A to trans-repress PR-B is weaker in the human lineage, suggesting the pregnancy maintenance functions of PR-B are stronger and the parturition inhibiting functions of PR-A are weaker in the human lineage than other species.

myometrium has been reported to be 0.5 (a PR-B dominant state) at 30-week gestation, 1.0 at term prior to the onset of labor, and 3.0 at term during labor (a PR-A dominant state) [67].

While isoform switching is ancestral in primates and rodents and thus did not evolve to initiate parturition, there is evidence supporting its role in the initiation of human labor [67,68,71–77]. Furthermore, the role of progesterone in the maintenance of pregnancy up to term is supported by studies showing that disruption of progesterone signaling by the PR-A/PR-B antagonist RU486 at any stage of pregnancy results in myometrial contractions and the initiation of labor in rodents [78–80] and primates [64], including humans [81]. Our observation that the human PR-A is a weaker trans-repressor of PR-B suggests that if isoform switching plays a role in the initiation or progression of parturition, this function may have changed in the human lineage. For example, if repression of PR-B mediated progesterone signaling by PR-A is important for initiating parturition at term, as proposed by the isoform switching hypothesis [68], then human PR-A may be less effective at inhibiting progesterone signaling and initiating parturition than PR-A of other species (**Fig 4**). We note, however, that PRs regulate the expression of genes without canonical progesterone response elements in their promoters, which may restrict the ability of PR-A to trans-repress PR-B.

## Conclusions, caveats, and limitations

We have shown that *PGR* evolved rapidly in the human stem-lineage, likely reflecting a continuing trend of relaxed purifying selection in Catarrhine primates, rather than positive selection. Regardless of the inference of positive or relaxed selection, we found that the AncHuman PR-A and PR-B diverged in their ability to trans-activate/trans-repress luciferase expression from the dPRL-332 promoter in mouse embryonic fibroblasts (MEFs). Potential limitations of our study include the use of the dPRL-332 promoter and MEFs instead of progesterone-responsive cell types. However, previous studies have shown that the functions of PR-A and PR-B are neither species, cell type, nor promoter-specific and are similar in transiently transfected cells, as well as in cells with a stably integrated reporter gene [7,54,82,83]. Therefore, it is likely that our results will be consistent in other cell types and promoter contexts. It is possible, however, that compensatory changes in *PGR* transcript expression compensate for changes in PR-A and PR-B protein function; this possibility is difficult to test without gene expression data from female reproductive tissues in other apes. Our observations suggest that the

regulation of progesterone signaling by PR-A and PR-B might have diverged in the human lineage, meaning that animal models of progesterone signaling may not adequately recapitulate human biology. Thus, the development of organoid models of the human maternal-fetal interface [84–88] will be important to understand human-specific aspects of progesterone signaling. Finally, while we have focused much of our discussion on how functional divergence of PR-A and PR-A relates to the isoform switching hypothesis, the progesterone receptor underlies multiple biological functions (such as decidualization), suggesting functional divergence may also affect those processes.

## Methods

### Identification of *PGRs* and evolutionary analyses

We identified *PGRs* from BLAST and BLAT searches of assembled and unassembled genomes. Short reads identified from unassembled genomes were assembled with the Geneious 'map to reference' option. *PGRs* were aligned with MAFFT (v7) using the auto (FFT-NS-1, FFT-NS-2, FFT-NS-i or L-INS-i) alignment option, 200PAM/k = 2 nucleotide scoring matrix, and 1.53 gap opening penalty. Alignment uncertainty was assessed by using GUIDANCE2, which constructed 400 alternative tree topologies by bootstrapping the multiple sequence alignment (MSA) generated by MAFFT (with the best fitting alignment algorithm, FFT-NS-i). Each bootstrap tree is used as a guide tree to re-align the original sequences, and alignment uncertainty inferred from differences in alignment among the bootstrap replicates. The overall GUIDANCE2 alignment score (0.988) was high, indicating MSA uncertainty was low. To ensure that poorly aligned regions of the MSA did not adversely affect downstream analyses, residues with an alignment quality less than 0.67 were masked to N. The alignment is provided in supplementary materials.

We first inferred the best fitting model of nucleotide substitution (012310) using the Datamonkey webserver [89]. To test if the $d_N/d_S$ ratio in the human stem-lineage was significantly different than in other lineages, we compared a model in which the $d_N/d_S$ ratio in the human stem-lineage was estimated separately from all other lineages (two rate model) to model with a single $d_N/d_S$ ratio for all lineages. This one rate and two rate comparison is implemented in the 'TestBranchDNDS.bf' module of HyPhy (2.22) [44]. Based on exploratory analyses, we allowed for variable $d_N$ and $d_S$ across sites with 3 rate classes. aBSREL [90], BUSTED [46], FEL [47], SLAC [47], FUBAR [48], MEME [49], and RELAX [50] were run using the Datamonkey webserver, whereas RELAX-scan was run locally using the developer version of HyPhy. We used the Datamonkey webserver and the best fitting model of nucleotide substitution (012310) to infer ancestral sequences. Ancestral human (AncHuman) and ancestral human-chimpanzee (AncHominini) genes were inferred with 1.0 Bayesian Posterior Probabilities. Phylogenetic tests of adaptive evolution (particularly branch-site methods) assume that nucleotide substitutions occur singly and independently. However, multinucleotide mutations (simultaneous mutations at two or three codon positions) can be common and cause false inferences of lineage-specific positive selection [91]. We note that human-specific amino acid substitutions result from single nucleotide mutations, G159R [GGG->CGG], A172V [GCT->GTT], S192A [TCC->GCC], G201E [GGG->GAG], P216A [CCC->GCC], S347C [TCT->TGT], P430T [CCC->ACC], and I662V [ATT->GTT], thus these results cannot be explained by the multinucleotide mutations bias.

### Cell culture and luciferase assay

We used mouse embryonic fibroblasts (Wt MEFs, ATCC CRL-2291) to assay PR trans-activation ability, to ensure that difference in trans-activation between ancestral PR proteins was not

because of trans-acting factors in human cells interacting more efficiently with AncHuman PR than AncHominini PR or the reverse. MEFs also do not express endogenous *Pgr*, ensuring that the luciferase expression from the dPRL-332 promoter was not affected by endogenous PR levels. MEFs were grown in maintenance medium containing Phenol Red-free DMEM [Gibco] with 10% charcoal-stripped FBS [Gibco], 1x insulin-transferrin-selenium [ITS; Gibco], 1% L-glutamine [Gibco] and 1% sodium pyruvate [Gibco]. 8000 cells were plated per well of a 96-well plate and 18 hours later cells in 80μl of Opti-MEM [Gibco] were transfected with 100ng of one or both of the PR plasmids, 100ng of dPRL-332 vector and 10ng of pRL-null vector with 0.1μl of PLUS Reagent and 0.25μl of Lipofectamine LTX [Invitrogen] in 20μl Opti-MEM. As a negative control, 100ng of dPRL-332 vector, 100ng of pcDNA3.1/V5-HisA vector and 10ng pRL-null vector were used. Cells were incubated with the transfection mixture for 6 hours. Then, cells were washed with DPBS [Gibco] and incubated in the maintenance medium overnight. The next day, decidualization medium (Phenol Red DMEM+GlutaMAX [Gibco] with 2% FBS [Gibco], 1% sodium pyruvate, 0.5mM 8-Br-cAMP [Sigma] and 1μM medroxy-progesterone acetate [MPA; Sigma]) was added and cells were incubated for 48 hours. Upon completion of the treatment, cells were washed with DPBS and incubated for 15 minutes in 1x Passive Lysis Buffer from Dual Luciferase Reporter Assay kit [Promega] with shaking. Luciferase and Renilla activities were measured on Glomax multi+ detection system [Promega] using Luciferin Reagent and Stop&Glow as described in the manufacturer's protocol. Experiment was repeated 4 times, each condition and control having 12 replicates. We standardized Luciferase activity values to Renilla activity values, and treatment to negative control. Effect size estimates (expressed as log mean difference between groups) and *P*-values were calculated using the DABEST package [Data Analysis with Bootstrap ESTimation] in R [92]. 5000 bootstrap samples were taken and the confidence interval is bias-corrected and accelerated. The *P*-value (s) reported are the likelihood[s] of observing the effect size(s), if the null hypothesis of zero difference is true.

## Supporting information

**S1 Table. Excel file with sheets that include Datamonkey output for each selection analysis and Luciferase data shown in Fig 3.**
(XLSX)

**S1 Data. PGR alignment file.**
(FASTA)

## Acknowledgments

The authors would like to thank the members of the Chicago-Northwestern-Duke March of Dimes Prematurity Research Center for comments on earlier versions of this work.

## Author Contributions

**Conceptualization:** Mirna Marinić, Vincent J. Lynch.

**Data curation:** Mirna Marinić, Vincent J. Lynch.

**Formal analysis:** Mirna Marinić, Vincent J. Lynch.

**Funding acquisition:** Vincent J. Lynch.

**Investigation:** Mirna Marinić, Vincent J. Lynch.

**Methodology:** Mirna Marinić, Vincent J. Lynch.

**Project administration:** Vincent J. Lynch.

**Resources:** Mirna Marinić, Vincent J. Lynch.

**Software:** Vincent J. Lynch.

**Supervision:** Vincent J. Lynch.

**Validation:** Mirna Marinić, Vincent J. Lynch.

**Visualization:** Vincent J. Lynch.

**Writing – original draft:** Mirna Marinić, Vincent J. Lynch.

**Writing – review & editing:** Mirna Marinić, Vincent J. Lynch.

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
