## [Decision Letter · Decision Letter 0]

2 Jan 2020

Dear Dr Lynch,

Thank you very much for submitting your Research Article entitled 'Relaxed constraint and functional divergence of the progesterone receptor (PGR) in the human stem-lineage' to PLOS Genetics. Your manuscript was fully evaluated at the editorial level and by independent peer reviewers. The reviewers appreciated the attention to an important topic but identified some aspects of the manuscript that should be improved.

We therefore ask you to modify the manuscript according to the review recommendations before we can consider your manuscript for acceptance. Your revisions should address the specific points made by each reviewer.

[LINK]

Yours sincerely,

Justin C. Fay

Associate Editor

PLOS Genetics

Bret Payseur

Section Editor: Evolution

PLOS Genetics

The reviews were quite positive but have identified some areas that need clarification and further discussion. In addition to responding to the reviewer comments it would be useful to better place your results into the context of those from Chen 2008. Chen 2008 identified the human and chimpanzee clade together as providing the strongest evidence for positive selection. It would thus make sense to determine whether you get the same results (no positive selection, just relaxed selection) if you consider the human/chimp clade as the focal point compared to just the human-stem lineage as you have already done.

Reviewer's Responses to Questions

**Comments to the Authors:**

Reviewer #1: Marinic and Lynch analyzed the molecular evolution of the human PGR gene by examining 119 Eutherian PGR sequences, including species and sub-species from each primate lineage, using multiple algorithms (shown in Table 1-4). Their findings suggest that PGR evolved under relatively strong purifying selection and that the human stem-lineage in particular shows evidence of rapid evolution. Their study resolves this issue which was uncertain based on previous analyses. Marinic and Lynch also examined the transcriptional activity of the ancestral human and human/chimp PR-A and PR-B isoforms using the human prolactin promotor driving a luciferase reporter in the context of mouse embryo fibroblast cells. Their results indicate that the ancestral human PR-B evolved to have stronger trans-activation whereas the transrepression of PR-A was less in the human PR-A than in the human/chimp PR-A. The authors conclude that the transcriptional activities of PR-A and PR-B may have diverged in the human lineage compared with non-human animals, although their data is limited to transcriptional activity based on the prolactin promoter in mouse cells. This is a well-written manuscript that brings important insight to understanding the distinctive actions of progesterone in human reproductive biology.

Critique:

1. Conclusions are skewed toward the isoform shifting hypothesis to explain how progesterone actions are withdrawn to trigger human parturition. This is based on PR activity at the prolactin promoter. However, PRs regulate the expression of multiple genes in reproductive cells that do not have a canonical PRE in their promotors. In this context the transrepressive activity of PR-A may not apply. This should be acknowledge in the discussion. Also the implication of the study to other distinctive actions of progesterone in human physiology (e.g., decidualization) could be discussed.

2. Figure 1: What dose 0.3 on the x-axis mean?

3. Figure 2: Please define the Y-axis; what is bits?

Reviewer #2: The authors of the manuscript “Relaxed constraint and functional divergence of the progesterone receptor (PGR) in the human stem-lineage” report divergent functions of the PR isoforms within the Catarrhine lineage that coincides with the detection of relaxed selection intensity. This report is significant given the role of progesterone in human reproduction and disease. Furthermore, our current understanding of the effects of the levels of progesterone and its receptors is limited.

Within the introduction the authors elaborate on the significance and context of PR. Overall the authors present a compelling argument for the importance of understanding the evolution of PR. There are some specific issues within the introduction, however, that should be addressed.

1. The authors lay out the features of PR that are critical to understanding their results—such as the trans-activation and trans-repression. While the manuscript focuses on the evolution and function of the PR gene, there is some evidence that regulation of PR is also important and the authors give examples in the conclusions (PR-A/PR-B abundance section). Could evolution of PR-mediated regulation be regulatory? Do these experiments address this possibility?

2. The logic of the paragraph beginning “While female reproductive” is difficult to follow. The first section states that well conserved traits can have underlying genes that evolve rapidly. The second section states that lineages with divergent traits therefore should also have rapidly evolving genes. Are all reproductive traits likely to have rapidly evolving underlying genes? It is unclear how these examples “suggest that PR may have evolved rapidly”. Furthermore, it is unclear how the citations “Arbeitman et al. 2002, Meiklejohn et al. 2003 and Parisi et al. 2003” on Drosophila expression and gene evolution are associated with Eutherian and Therian reproductive gene evolution.

3. The authors suggest that Catarrhine primates have “a divergent (and unknown) parturition signal” and yet in the discussion discuss at length the evidence for PR as a mechanism of parturition. The exact and entire mechanism is unknown but there is a large body of research into parturition.

4. The authors thoroughly and insightfully cover the evidence for selection on PR along the human lineage. However, they do not discuss recent selection among human populations (Li et al 2018; “Natural Selection Has Differentiated the Progesterone Receptor among Human Populations”). How does evidence of both positive and balancing selection in recent human history support or contradict the findings in this manuscript?

The authors set about to show that 1) PGR evolved rapidly in the human lineage; 2) Rate acceleration is consistent with relaxed purifying selection and not positive selection; and 3) Relaxed selection resulted in altered PR function. The methods used to test these hypotheses are robust and appropriately applied.

1. The authors use HyPhy to test for rapid evolution along the human lineage. The sentence starting “The two w model was significantly better…” is confusing. First, w2 is not defined (or detailed in the supplements) but I assumed it is associated with the stem-lineage (Figure 1 asterisk). Second, it is not clearly laid out that this is the evidence for rapid evolution along the stem lineage. The authors may also want to define “rapid evolution” as an increase in the rate of nucleotide substitution compared to other lineages (as it is defined in human accelerated regions in Pollard 2006) to avoid confusion between absolute substitution rate and rate of evolution.

2. The authors present multiple lines of evidence to suggest that positive selection is not responsible for the rapid evolution of PRG. They account for multiple models of positive selection including episodic and weak selection. These results are reported in Tables 1 – 4. It would be useful to distinguish sites where there is significant evidence that w < 1 versus sites where there is no evidence that w doesn’t = 1. This would be helpful since each of the methods has different abbreviations and methods for assessing significance. The authors also provide support the conclusion that relaxed selection is responsible for the elevated rate of evolution in PGR. This is represented in Figure 1. From the methods section and description, it is unclear if the phylogeny in Figure 1 is a species phylogeny (as suggested by the title “Phylogeny of species”) or a gene tree (only information from the PGR gene).

3. The authors also present both computational and experimental evidence that human specific changes in PGR are functionally important. The computational results using CADD suggest that the human-specific substitutions are deleterious. The likelihood of functional consequences from these changes is further supported by luciferase assays. Figure 3D would be strengthened if the p-values were reported in the figure or legend.

Overall the methods presented in the manuscript strongly support the conclusions of the authors. The only concern is the presentation of the evidence for “rapid evolution.” The authors demonstrate how the use of multiple measures/tools can help more precisely differentiate between modes of selection that are often difficult to distinguish.

The supplemental excel data covers almost all of the experiments conducted except the TestBranchDNDS data and the raw luciferase data. The authors should also consider providing the PGR sequences they identified and/or assembled.

In the discussion the authors summarize their results and place them in the context of PRs role in parturition. Most importantly the authors discuss how changes in repressor strength in PR-A may be involved in the differences between human parturition and that of common model organisms such as rodents and even close relatives such as chimps. The manuscript is well written but should be clarified in a few locations noted above. It is striking how little we know about human parturition and pregnancy—this manuscript meaningfully contributes to improving our understanding of these important topics.

**Have all data underlying the figures and results presented in the manuscript been provided?**

Reviewer #1: Yes

Reviewer #2: No: The supplemental excel data covers almost all of the experiments conducted except the TestBranchDNDS data and the raw luciferase data. The authors should also consider providing the PGR sequences they identified and/or assembled.

PLOS authors have the option to publish the peer review history of their article (what does this mean?). If published, this will include your full peer review and any attached files.

Reviewer #1: No

Reviewer #2: No

---

## [Editor Report · Decision Letter 1]

13 Feb 2020

Dear Dr Lynch,

We are pleased to inform you that your manuscript entitled "Relaxed constraint and functional divergence of the progesterone receptor (PGR) in the human stem-lineage" has been editorially accepted for publication in PLOS Genetics. Congratulations!

Yours sincerely,

Justin C. Fay

Associate Editor

PLOS Genetics

Bret Payseur

Section Editor: Evolution

PLOS Genetics

Comments from the reviewers (if applicable):

**Data Deposition**

http://datadryad.org/submit?journalID=pgenetics&manu=PGENETICS-D-19-01846R1

**Press Queries**

---

## [Editor Report · Acceptance letter]

10 Apr 2020

PGENETICS-D-19-01846R1 

Relaxed constraint and functional divergence of the progesterone receptor (PGR) in the human stem-lineage 

Dear Dr Lynch, 

We are pleased to inform you that your manuscript entitled "Relaxed constraint and functional divergence of the progesterone receptor (PGR) in the human stem-lineage" has been formally accepted for publication in PLOS Genetics! Your manuscript is now with our production department and you will be notified of the publication date in due course.

With kind regards,

Kaitlin Butler

PLOS Genetics

On behalf of:
